# Revisiting flow augmentation bypass for cerebrovascular atherosclerotic vaso-occlusive disease: Single-surgeon series and review of the literature

Jihad Abdelgadir[1]*, Aden P. Haskell-Mendoza[1], Amanda R. Magno[1], Alexander D. Suarez[1], Prince Antwi[1], Alankrita Raghavan[1], Patricia Nelson[1], Lexie Zidanyue Yang[2], Sin-Ho Jung[2], Ali R. Zomorodi[1]

1 Department of Neurosurgery, Duke University Medical Center, Durham, North Carolina, United States of America, 2 Department of Biostatistics and Bioinformatics, Duke University School of Medicine, Durham, North Carolina, United States of America

* jihad.abdelgadir@duke.edu

**Data Availability Statement:** All relevant data are within the paper and its Supporting information files.

## Abstract

### Objective

Despite advances in the nonsurgical management of cerebrovascular atherosclerotic steno-occlusive disease, approximately 15–20% of patients remain at high risk for recurrent ischemia. The benefit of revascularization with flow augmentation bypass has been demonstrated in studies of Moyamoya vasculopathy. Unfortunately, there are mixed results for the use of flow augmentation in atherosclerotic cerebrovascular disease. We conducted a study to examine the efficacy and long term outcomes of superficial temporal artery to middle cerebral artery (STA-MCA) bypass in patients with recurrent ischemia despite optimal medical management.

### Methods

A single-institution retrospective review of patients receiving flow augmentation bypass from 2013–2021 was conducted. Patients with non-Moyamoya vaso-occlusive disease (VOD) who had continued ischemic symptoms or strokes despite best medical management were included. The primary outcome was time to post-operative stroke. Time from cerebrovascular accident to surgery, complications, imaging results, and modified Rankin Scale (mRS) scores were aggregated.

### Results

Twenty patients met inclusion criteria. The median time from cerebrovascular accident to surgery was 87 (28–105.0) days. Only one patient (5%) had a stroke at 66 days post-op. One (5%) patient had a post-operative scalp infection, while 3 (15%) developed post-operative seizures. All 20 (100%) bypasses remained patent at follow-up. The median mRS score at follow up was significantly improved from presentation from 2.5 (1–3) to 1 (0–2), P = .013.

**Funding:** The author(s) received no specific funding for this work.

**Competing interests:** The authors have declared that no competing interests exist.

## Conclusions

For patients with high-risk non-Moyamoya VOD who have failed optimal medical therapy, contemporary approaches to flow augmentation with STA-MCA bypass may prevent future ischemic events with a low complication rate.

## Introduction

In the United States, approximately 795,000 individuals develop a stroke each year, of which 185,000 are recurrent [1]. A further 160,264 die from cerebrovascular disease, the 5th leading cause of mortality in 2020 [2]. Eighty-seven percent of strokes are ischemic, with the remainder consisting of intracerebral (10%) and subarachnoid hemorrhage (3%) [3].

Multiple randomized-controlled trials have generated high quality evidence for the use of antiplatelet therapy, anticoagulation, and thrombolysis for treatment and prevention of stroke [1]. Unfortunately, certain stroke etiologies, including severe intracranial large artery occlusion and patients with chronic retinal ischemia or limb-shaking transient ischemic attacks (TIAs), have a 15–20% risk of recurrence despite optimal nonsurgical therapy [4–9]. These recurrent ischemic events reduce patient quality of life and carry significant morbidity and mortality [1, 4]. In this subgroup of patients at high-risk for recurrent ischemia and functional loss, there remains a potential role for extracranial-intracranial (EC-IC) bypass.

Yaşargil performed the first superficial temporal artery to middle cerebral artery (STA-MCA) anastomosis in Zurich, Switzerland in 1967 to treat internal carotid artery (ICA) occlusion [6]. Following its development, EC-IC bypass was adopted the treatment of ischemic cerebrovascular disease, with publication of several retrospective series [6, 10–15]. This culminated in the International EC-IC Bypass trial, which tested whether the procedure in addition to best medical therapy (BMT) was superior to BMT alone in a heterogenous group of patients with ischemic cerebrovascular disease and ultimately demonstrated no difference in the incidence of fatal and nonfatal ischemic strokes between groups [16].

While a reduction in the use of STA-MCA bypass followed the trial's publication, critics noted that no hemodynamic criteria were used to stratify patients for inclusion, forming the basis for the Carotid Occlusion Surgery Study (COSS). Published in 2011, COSS randomized 195 patients with symptomatic internal carotid artery occlusion and cerebral ischemia on PET scans to bypass and BMT (n = 97) versus BMT alone (n = 98) and was stopped early for futility at 2 years [17]. 21% of patients in the surgical group developed ipsilateral stroke at the time of study termination versus 22.7% in the medical group; the perioperative stroke rates were 14.4% versus 2.0%, respectively. Requirements for participation in COSS included a 2-day workshop on microvascular anastamosis or at least 10 previous bypass cases, with supervision if below case thresholds [18]. There were no certification requirements for neuroanesthesia, neuro-intensive care, or nursing staff, suggesting a role for interventions to lower perioperative morbidity given higher than expected perioperative stroke rates [4, 17].

In the interval since these trials, indications for STA-MCA bypass have included complex intracranial aneurysms, skull base tumors with vascular involvement, and flow augmentation in Moyamoya disease [4]. There remains speculation that the aforementioned subsets of patients not included in either COSS or the EC-IC Bypass trial may benefit from surgical intervention [4–6]. In the post-COSS era, there has been mounting evidence for bypass in high-risk patients at centers with significant procedural experience [19–21]. Haynes and colleagues recently published a series of 8 patients treated with STA-MCA bypass between 2016–2019

following recurrent or rapidly progressive strokes despite optimal medical or endovascular treatment [21]. In this study, 88% of patients had no recurrent strokes, and 75% demonstrated functional improvement as measured by the modified Rankin scale (mRS). Similarly, our institution has continued to offer STA-MCA bypass for patients with symptomatic disease (i.e. recurrent strokes or "crescendo" ischemic symptoms) despite optimal medical treatment. Indeed, it has been our experience that this select group of patients with vaso-occlusive disease (VOD) derives benefit from operative intervention. We report a single-surgeon series of 20 patients receiving STA-MCA bypass for high-risk, symptomatic VOD with functional improvement and extremely low post-operative stroke rates.

## Materials and methods

### Patient selection and operation

For consideration of bypass, patients had to meet the following criteria: failure of best medical therapy (i.e. aspirin or dual antiplatelet therapy), defined as continued "crescendo" ischemic symptoms or strokes despite optimal nonsurgical management. All patients receive a CT head and neck angiogram or digital subtraction angiogram to assess internal and external carotid circulation and identify a possible donor and receipient vessel. Once in the operating room, the senior author's typical practice is to conduct Doppler ultrasound to identify the path of the superficial temporal artery. The resultant linear or curvilinear incision is made following the artery to expose 8–10 cm of the donor vessel only, with minimal additional dissection to expose the parietal branch of the STA. An arterial line is placed by neuroanesthesia for dedicated blood pressure management. Intraoperatively, patients are maintained at a mean arterial pressure (MAP) of 100. A recipient site requiring no sacrifice or manipulation of microcortical vessels is selected. The typical vessel clamp time during the bypass procedure is under 15 minutes. Bypass patency is confirmed intra-operatively with indocyanine green videoangiography and post-operatively via CT angiogram. All patients are cared for in our institution's dedicated neuro-intensive care unit following the operation. The arterial line remains in place with a goal MAP of 80–90 for the first 24 hours post-operatively.

### Data collection

This study was conducted in accordance with a protocol approved by the health system institutional review board (IRB # Pro00108340). A retrospective review of patients receiving STA-MCA bypass performed by a single surgeon (A.R.Z) from July 2013 to January 2021 was completed. Patients were included based on receipt of STA-MCA bypass for VOD. Patients who were undergoing STA-MCA bypass for Moyamoya disease, vertebrobasilar insufficiency, or aneurysm were excluded from the study. Other donor-recipient vessel pairs and bypasses for tumors were similarly excluded.

The primary outcome was time to post-operative stroke, confirmed as new areas of diffusion restriction on MRI on the ipsilateral side of the bypass. Additional measures included patient demographics and comorbidities, CT perfusion findings preceding and following surgery, imaging characteristics of post-operative stroke, functional outcomes as measured by the modified Rankin scale (mRS) at discharge and follow-up, and peri- and post-operative complications.

### Statistical analyses

For patients with atheroscleroticVOD, continuous variables were summarized with means, standard deviations, medians, interquartile ranges, and ranges. Categorical variables were

summarized with frequency counts and percentages. Patients were censored at the last date of follow up. The statistical significance level was set at P = 0.05. All tests were two-sided. These analyses were performed using SAS version 9.4 (SAS Institute Inc., Cary, NC). The Kaplan-Meier method was used to estimate the post-operative stroke-free probability and draw the stroke-free survival curve. Change in mRS from time of presentation to discharge or follow-up was compared using Wilcoxon matched pairs signed rank tests in GraphPad Prism (GraphPad Inc., San Diego, CA).

## Results

### Patient clinical and surgical characteristics

During the study period, 79 patients were treated with EC-IC bypass. Of those excluded, 39 (49.4%) received bypass for Moyamoya disease and 20 (25.3%) for aneurysm or vertebrobasilar insufficiency. The remaining 20 patients received open vascular surgery for atherosclerotic VOD. Baseline characteristics for the study cohort are shown in Table 1. The median age in our study was 64 years (range 45–78); 10 (50.0%) patients identified as female. Nine (45.0%) patients were former smokers, and a further 4 (20.0%) were current smokers. The median modified Rankin scale (mRS) score at the time of presentation was 2.5 (interquartile range (IQR) 1–3), with the last cerebrovascular accident occurring at a median of 87 days (IQR 28–1195 days) prior to surgery (Table 2 & Fig 1A).

We next reviewed baseline medications and comorbidities (Table 1). All patients were receiving aspirin at presentation. A further 14 (70.0%) were receiving dual-antiplatelet therapy (DAPT), and 6 (30.0%) patients were on anticoagulation. Comorbidities included hypertension in 17 (85.0%) patients, hyperlipidemia in 10 patients (50.0%), diabetes mellitus in 8 patients (40.0%), renal disease in 5 patients (25.0%), and atrial fibrillation in 1 patient.

### Cerebrovascular outcomes following STA-MCA bypass for VOD

The median duration of follow-up during the study period was 213.5 days (36–1330 days). The primary outcome of stroke-free survival was analyzed using the Kaplan-Meier method and is shown in Fig 1B. Remarkably, only 1 (5%) of the 20 patients suffered from post-operative stroke at 66 days.

We additionally analyzed pre- and post-operative CT imaging for cerebrovascular abnormalities, shown in Table 3. Only 8 (40.0%) patients had pre-operative CT perfusion studies available for analysis, all of which were abnormal. A subset of 6 patients had matched post-operative CT perfusions. For all patients, post-operative imaging showed improved perfusion.

Pre- and post-operative MRIs were reviewed for stroke (Table 4). Pre-operative MRIs demonstrated evidence of infarction in 9 (45.0%) patients. Only one patient (5%) had postoperative stroke on MRI with involvement of the right-sided MCA territory.

Functional status as measured by mRS was also improved following treatment in our cohort (Fig 1A). The median mRS at discharge was 2 (IQR 1.25–2.75), with median mRS at the time of follow-up decreased to 1, no significant disability (IQR 0–2). We found that the median mRS at follow-up was significantly improved from initial presentation (1 vs 2.5, P = 0.013) and from discharge (1 vs 2, P = 0.0002), but was not significant in the early period from initial presentation to discharge (P = 0.85), suggesting an influence of early post-operative debility on functional status.

**Table 1. Baseline patient characteristics.**

|  | VOD (N = 20) |
| --- | --- |
| **Age** | - |
| Mean (SD) | 61.9 (9.8) |
| Median | 64.0 |
| Q1, Q3 | 52.5, 70.0 |
| Range | (45.0–78.0) |
| **Sex** | - |
| Female | 10 (50.0%) |
| Male | 10 (50.0%) |
| **Smoking Status** | - |
| Never | 7 (35.0%) |
| Current | 4 (20.0%) |
| Past | 9 (45.0%) |
| Unknown | 0 (0.0%) |
| **Vascular Territory**[a] | - |
| ACA | 6 (30.0%) |
| MCA | 17 (85.0%) |
| PCA | 4 (20.0%) |
| **Hemoglobin-A1c** | - |
| Mean (SD) | 8.0 (2.8) |
| Median | 6.4 |
| Q1, Q3 | 5.8, 11.2 |
| Range | (5.2–12.6) |
| **Aspirin** | - |
| Yes | 20 (100.0%) |
| **Dose of Aspirin (mg)** | - |
| 81 | 12 (60.0%) |
| 325 | 8 (40.0%) |
| **Dual antiplatelet therapy** | - |
| No | 6 (30.0%) |
| Yes | 14 (70.0%) |
| **Anticoagulation** | - |
| No | 14 (70.0%) |
| Yes | 6 (30.0%) |
| **Comorbidities** | - |
| HTN | 17 (85.0%) |
| DM | 8 (40.0%) |
| HLD | 10 (50.0%) |
| Renal Disease | 5 (25.0%) |
| Atrial Fibrillation | 1 (5.0%) |

**Abbreviations:** A1C, hemoglobin A1C; ACA, anterior cerebral artery; DM, diabetes mellitus; HLD, hyperlipidemia; HTN, hypertension; MCA, middle cerebral artery; PCA, posterior cerebral artery; SD, standard deviation.
[a]Categories not mutually exclusive.

## Complications related to STA-MCA bypass

Although the risk of stroke was low in the study cohort, we examined rates of additional post-operative complications in our sample (Table 5). No patients suffered from hemorrhage. One

**Table 2. Surgical characteristics of patients receiving STA-MCA bypass for atherosclerotic vaso-occlusive disease (VOD).**

|  | VOD (N = 20) |
| --- | --- |
| **Days from Last Cerebrovascular Accident (CVA) to Surgery** | - |
| Mean (SD) | 188.1 (367.1) |
| Median | 87.0 |
| Q1, Q3 | 28.0, 195.0 |
| Range | (0.0–1696.0) |
| **Surgery** | - |
| Left bypass | 12 (60.0%) |
| Right bypass | 8 (40.0%) |
| **Type of Bypass** | - |
| Single barrel | 19 (95.0%) |
| Double barrel | 1 (5.0%) |

**Abbreviations:** SD, standard deviation.

Most patients presented with disease attributable to the anterior circulation (16 patients, 80.0%), with the MCA being the most common site (17, 85.0%) (Table 2). Twelve (60.0%) patients subsequently underwent left-sided bypass, with 8 receiving right-sided surgery. Almost all patients (19, 95.0%) received a single barrel bypass, with 1 receiving double barrel bypass.

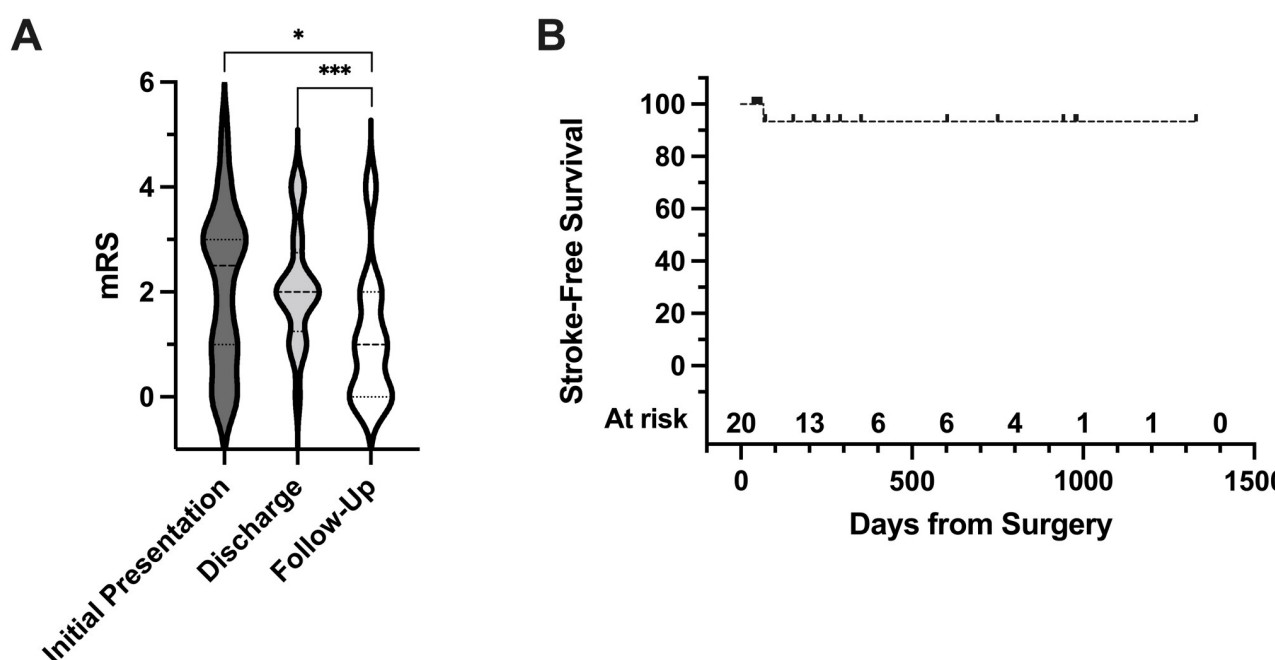

**Fig 1. STA-MCA bypass for VOD is associated with improvement in post-operative median Modified Rankin Scale score and lowered stroke risk. A.** The Modified Rankin Scale (mRS) score at the time of presentation, discharge, and follow up was collected for all patients (n = 20) and shown on violin plots. Medians are shown as dashed lines, with the interquartile range represented as dotted lines. The Wilcoxon matched-pairs signed rank test was used to make non-parametric comparisons between group medians. P = .013 for initial presentation [median mRS 2.5 (IQR 1–3)] to follow-up [1 (0–2)] and P = .0002 for discharge [2 (1.25–2.75)] to follow-up, P = .85 for initial presentation to discharge, not significant. **B.** Kaplan-Meier curve of stroke-free survival following receipt of STA-MCA bypass for VOD (n = 20). Median not reached. One patient had a stroke at post-operative day 66.

**Table 3. Pre- and postoperative CT perfusion findings for patients receiving STA-MCA bypass.**

| | VOD (N = 20) |
|---|---|
| **Pre-operative CT Perfusion**[a] | |
| ACA Oligemia | 5 (25.0%) |
| MCA Oligemia | 8 (40.0%) |
| PCA Oligemia | 2 (10.0%) |
| Unavailable | 12 (60.0%) |
| **Post-operative CT Perfusion** | |
| Unavailable or no comparator | 14 (70.0%) |
| Improved | 6 (30.0%) |

[a]Categories are not mutually exclusive.

**Table 4. Pre- and postoperative MRI findings.**

| | VOD (N = 20) |
|---|---|
| **Pre-operative Stroke Present on MRI*** | 9 (45.0%) |
| ACA Territory | 2 (10.0%) |
| MCA Territory | 8 (40.0%) |
| PCA Territory | 1 (5.0%) |
| **Post-Op Stroke Present on MRI**[a] | 1 (5.0%) |
| ACA Territory | 0 (0.0%) |
| MCA Territory | 1 (5.0%) |
| Ipsilateral side of bypass? | 1 (5.0%) |
| PCA Territory | 0 (0.0%) |

[a]Categories are not mutually exclusive.

patient developed a post-operative scalp infection. Long term bypass patency was excellent, with no occlusion post-operatively. Seizures were noted in 3 of 20 patients (15.0%).

## Comparison to published studies

We reviewed the literature for post-COSS series of EC-IC bypass for steno-occlusive disease and compared them to our own as shown in Table 6 [7, 8, 19–24]. Seven of 8 studies (87.5%) were retrospective in nature. Excluding a single case report, the number of patients treated ranged from 12 to 179, with a mean cohort age of 55 to 62 years. Treatment criteria were qualitatively broad, but stringent, often focusing on medically refractory or recurrent ischemic symptoms and incorporating a variety of diagnostic testing (acetazolamide challenge, CT

**Table 5. Perioperative and postoperative complications associated with STA-MCA bypass for vaso-occlusive disease.**

| | VOD (N = 20) |
|---|---|
| **Complications** | - |
| Hemorrhage | 0 (0.0%) |
| Infection | 1 (5.0%) |
| Bypass occlusion | 0 (0%) |
| Stroke | 1 (5.0%) |
| Seizure | 3 (15.0%) |

**Table 6. Comparison and review of post-COSS literature on direct bypass for VOD.**

| Author | Year | Study type | Number of patients | Mean age | Criteria for Intervention | Post-operative mRS score | Perioperative stroke rate | Perioperative hemorrhage | Bypass patency | Notes |
|---|---|---|---|---|---|---|---|---|---|---|
| Kuroda et al. [23] | 2014 | Prospective, single-arm | 25 | 62.9 ± 11.0 | Severe ICA or MCA (90%) occlusion, no or small infarct on MRI, Type 3 ischemia and elevated oxygen extraction fraction | - | 1 (4%) | 0 (0%) | - | "Double" bypass—frontal and parietal STA anastomosed to MCA. |
| White et al. [19] | 2019 | Retrospective, case series | 35 | 55 (22–74) | Carotid or MCA stenosis with failure of optimal medical therapy (recurrent strokes or TIAs), or perfusion-dependent neurological exam | - | 3 (8.6%) | 0 (0%) | 33 (94%) | - |
| Steinberg et al. [20] | 2020 | Retrospective, case series | 17 | 62 ± 11 | Progressive ischemic symptoms (TIA, misery perfusion), ongoing ischemic penumbra on MRI or CTP despite medical management | 13 (85%) patients ≤ 2 | 3 (17.6%) | 3 (17.6%) | 17 (100%) | - |
| Haynes et al. [21] | 2021 | Retrospective, case series | 8 | 60 ± 6 | Symptomatic recurrent or rapidly progressive stroke or TIA with hypoperfusion despite optimal medical management or endovascular therapy | Median 1 (IQR 0–3) | 0 (0%) | 0 (0%) | 8 (100%) | One patient did not recover from presenting stroke and expired 4 months post-op due to bilateral strokes |
| Wessels et al. [8] | 2021 | Retrospective study | 179 (186 total bypasses) | 58 ± 12 | Symptomatic VOD with recurrent TIAs or stroke under best medical management with impaired cerebrovascular reserve (≥ 30% reduction in baseline perfusion of affected territory during acetazolamide challenge) | - | 8 (4.3%) | 3 (1.6%) | 175 (94%) | Patients stratified as atherosclerotic ICA occlusion vs atherosclerotic multivessel disease |
| Aono et al. [24] | 2021 | Case report | 1 | 69 | Left ICA occlusion with TIA and multiple aneurysms on DAPT | - | 0 (0%) | - | 1 (100%) | STA and posterior auricular artery-MCA bypass. Improvement of average Wechsler Adult Intelligence score from 71 to 89.25 |
| Housley et al. [22] | 2022 | Retrospective, case series | 27 | 59.9 ± 10.1 | Medical optimization, diagnosis of compromised or impaired cerebrovascular reserve | 17 (63%) mRS 1 to 2 at last follow-up | 2 (6.3%) | - | 30 of 32 bypasses (93.8%) | Focus on distal internal carotid or proximal MCA stenosis, multiple EC-IC bypass techniques |

*(Continued)*

**Table 6.** (Continued)

| Author | Year | Study type | Number of patients | Mean age | Criteria for Intervention | Post-operative mRS score | Perioperative stroke rate | Perioperative hemorrhage | Bypass patency | Notes |
|--------|------|-----------|-------------------|----------|---------------------------|--------------------------|---------------------------|--------------------------|----------------|-------|
| Zhao et al. [7] | 2022 | Retrospective, case series | 12 | 55.8 ± 6.7 | Patients with ICA occlusion with > 2 DWI-MRI proven strokes or TIAs in 6 months on best medical therapy with mRS ≤ 3, 40–70 years, and no severe heart disease | - | 1 (8.3%) | 1 (8.3%) | 11 (91.7%) | Double bypass, Post-op median modified Barthel Index 82.5 (IQR 75–90 from 41 (35–50.25) |

perfusion, MRI). Only 3 studies other than our own reported functional outcomes. The perioperative stroke rate ranged from 4.3% to 17.6% for a study focusing on "rescue bypass" for evolving ischemia. Bypass patency rates for all reported studies were above 90%.

## Discussion

The STA-MCA bypass is a versatile open vascular intervention for flow diversion and augmentation that has fallen out of favor in non-Moyamoya VOD as a result of failure to demonstrate benefit in two large trials [16, 17]. In the interim, stroke and TIA outcomes have greatly improved due to advances in medical and endovascular therapies. However, there remains a subset of patients with high-risk cerebrovascular disease who go on to have recurrent ischemic strokes that carry significant morbidity and mortality. We have demonstrated that with expert surgical and anesthesia care, STA-MCA bypass significantly improved the mRS by -1.5 points from the time of presentation to follow-up for this selected group. Although the cohort was small, only 1 patient (5%) had a post-operative stroke, similar to the medical management arms of COSS (2.0%, ipsilateral ischemic stroke) and SAMMPRIS (5.8%) as well as to other post-COSS evaluations of bypass (Table 6) [9, 17]. Indeed, a recent systematic review of EC-IC bypass for VOD suggested a trend towards decreased perioperative (5.7%) and overall stroke (9.1%) rates over time [25]. The majority had received evidence-based medical therapy with aspirin (100%) or DAPT (70.0%) and had comorbidities commensurate with increased stroke risk. These data suggest that STA-MCA bypass should remain an option for patients with recurrent strokes who fail medical therapy.

The EC-IC Bypass and COSS trials are landmark accomplishments within neurosurgery. However, neither trial required patients to have developed recurrent stroke prior to entry. Specific presentations, such as crescendo or limb-shaking TIAs, chronic retinal ischemia, and severe intracranial large artery occlusion represent roughly 10% of ischemic strokes and are associated with a high risk of recurrence of 15–20% in the setting of medical therapy [4–6, 8, 9]. Indeed, a retrospective analysis of 179 German patients during 2012–2019, contemporaneous with our study, suggested the current population referred for bypass for VOD had frequent multivessel disease (52%), recurrent ischemic symptoms (80%), and greater comorbidities [8]. Our study provided evidence of improved functional outcomes following bypass in one of the largest post-COSS series of patients with recurrent ischemic disease. This benefit has been corroborated by other centers [7, 19–22]. In addition to work by Haynes and colleagues, Zhao et al. showed improved imaging and cognitive outcomes in 12 patients with recurrent strokes due to intracranial large artery occlusion, with only one perioperative TIA [7]. White et al. demonstrated excellent graft patency of 94% with only 3 perioperative strokes in an unselected

subset of 35 patients with symptomatic VOD treated with bypass after publication of COSS [19]. Further, Steinberg et al. reported a role for rescue bypass for 17 patients acutely presenting with refractory or progressive VOD, with 85% of patients achieving a mRS score of ≤ 2 over 10 months of follow-up [20]. A recently published study further demonstrated the safety of direct EC-IC bypass using a variety of techniques, with a perioperative stroke rate of 6.3% [22]. Taken together, the published studies demonstrate the safety and feasibility of bypass for patients with complex or high-risk disease.

Similar to the above series and post-hoc analyses of the COSS trial, our bypass patency rate was extremely high, with imaging displaying evidence of improved perfusion [18]. While this surgical result is remarkable at face-value, the question remains regarding the etiology of perioperative stroke in the original trials of EC-IC bypass if unrelated to the operation itself. Compared to these RCTs, our study's non-stroke complications (3 patients with seizures and 1 perioperative infection) are commensurate with those reported. Further, given the single-surgeon nature of this study, technical factors associated with the surgery are controlled, whereas both COSS and the EC-IC Bypass trials had limited standardization of anesthesia, neuro-intensive care, and nursing [4, 5]. Our outcomes suggest that standardization of perioperative management post-COSS mitigate the risk of stroke. Data from two additional trials may identify risk factors for perioperative stroke and provide evidence supporting bypass for atherosclerotic VOD. The Japanese EC-IC Bypass Trial (JET) was a multicenter RCT assessing the role of STA-MCA bypass plus BMT versus BMT alone in patients with reduced cerebral blood flow on single-photon emission computed tomography from chronic ICA or MCA occlusion [4, 23]. One-hundred and ninety-six patients were randomized 50:50 to each arm, with interim analyses showing a statistically significant reduction in the primary outcome of major stroke and death at interim analysis (5.1% vs. 14.3% for surgically-treated vs medically-treated patients) [26]. Unfortunately, this study has not yet been published in an English-language journal, precluding detailed analysis of perioperative stroke or complication risk [23, 26, 27]. The Carotid and Middle Cerebral Artery Occlusion Surgery Study (CMOSS) was conducted in China with planned randomization of 330 patients with ICA or MCA occlusion and hemodynamic insufficiency to EC-IC bypass with BMT versus BMT alone; it has been completed as of March 2020 (NCT01758614) [28]. The primary study outcome was stroke from randomization to 30 days post-operatively and ipsilateral ischemic stroke within 2 years. Given the importance of assessing perioperative risk, the results of CMOSS are anxiously awaited. Future reports should address how perioperative risks might be mitigated and standardized to prevent early stroke.

Our study demonstrated functional improvement as measured by mRS score. While single-point changes in the mRS score are clinically relevant, the seven-level ordinal scale has been subjected to dichotomous analyses throughout its use as an endpoint in stroke RCTs; further, repeated measures of the mRS score highlight a general increase due to post-stroke recovery irrespective of treatment [29, 30]. Concerns about reproducibility and increased emphasis on patient-centered outcomes and quality of life have resulted in increased use of adjunct measures [29]. An important ancillary study to COSS was the Randomized Evaluation of Carotid Occlusion and Neurocognition (RECON) trial, which hoped to identify whether patients receiving bypass in COSS had improved or preserved neurocognition at 2 years. Unfortunately, this study was not completed due to termination of COSS, though analyses of the 29 (13 surgical, 16 medical) patients remaining at the 2-year endpoint showed no difference in cognitive change between arms when controlling for age, education, and depression [31]. However, there are reports of cognitive improvement following EC-IC bypass on neuropsychiatric testing [24]. Likewise, assessment of follow-up mRS and the Barthel Index are planned

for inclusion in CMOSS [28]. These findings hint at a role for assessment of cognitive and quality of life outcomes following bypass in the post-COSS era.

Further, direct STA-MCA bypass may not be the only option with benefit in high-risk vaso-occlusive disease [32]. While indirect bypass methods were previously thought to be ineffective for non-Moyamoya VOD, the recently published results of the phase II Encephaloduroarterio-synangiosis Revascularization for Symptomatic Intracranial Arterial Stenosis (ERSIAS) trial showed a rate of 9.6% (5 of 52 patients) for the composite primary endpoint of 30-day postoperative stroke or death or stroke in the territory of the bypassed artery beyond 30 days [33]. Further studies are warranted identify which patients or subgroups have high-risk disease phenotypes that will be most responsive to any or all of these interventions.

## Limitations

This study represents one of the largest post-COSS series of STA-MCA bypass for non-Moyamoya atherosclerotic VOD, but is limited by its small sample size, single-center and retrospective nature. Given the small population of patients with each of the individual recurrent stroke phenotypes above, we instead analyzed this population in aggregate. The median follow-up duration in this study was 7.7 (1.2–43.7) months. This study was conducted at a tertiary referral center with a large catchment area, with a combination of location and patient specific factors resulting in limited follow-up. Nonetheless, this study represents an important contribution to the growing body of post-COSS literature of bypass for VOD following failure of medical therapy in high-risk stroke subtypes.

## Conclusions

STA-MCA bypass is a time-honored procedure that has utility in a variety of neurosurgical settings. Rigorous trials have identified a risk of perioperative stroke following bypass for non-Moyamoya cerebrovascular disease in non-refractory populations. We demonstrate a very low stroke risk in one of the largest cohorts since publication of these trials, with improvement in postoperative functional outcomes. For patients who have recurrent strokes despite maximal medical therapy, referral for bypass at centers of expertise may be beneficial.

## Supporting information

**S1 File. Minimal data set.** Contains de-identified underlying data used to conduct the study. (XLSX)

## Author Contributions

**Conceptualization:** Jihad Abdelgadir, Ali R. Zomorodi.

**Data curation:** Jihad Abdelgadir, Aden P. Haskell-Mendoza, Amanda R. Magno, Alexander D. Suarez, Prince Antwi, Ali R. Zomorodi.

**Formal analysis:** Aden P. Haskell-Mendoza, Lexie Zidanyue Yang, Sin-Ho Jung.

**Investigation:** Jihad Abdelgadir, Aden P. Haskell-Mendoza, Amanda R. Magno, Sin-Ho Jung.

**Methodology:** Jihad Abdelgadir, Sin-Ho Jung, Ali R. Zomorodi.

**Project administration:** Jihad Abdelgadir, Ali R. Zomorodi.

**Software:** Aden P. Haskell-Mendoza, Amanda R. Magno, Lexie Zidanyue Yang, Sin-Ho Jung.

**Supervision:** Jihad Abdelgadir, Sin-Ho Jung, Ali R. Zomorodi.

**Validation:** Lexie Zidanyue Yang.

**Visualization:** Aden P. Haskell-Mendoza, Lexie Zidanyue Yang.

**Writing – original draft:** Aden P. Haskell-Mendoza.

**Writing – review & editing:** Jihad Abdelgadir, Aden P. Haskell-Mendoza, Amanda R. Magno, Alexander D. Suarez, Prince Antwi, Alankrita Raghavan, Patricia Nelson, Lexie Zidanyue Yang, Sin-Ho Jung, Ali R. Zomorodi.

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
