## [Decision Letter · Decision Letter 0]

18 Apr 2023

PONE-D-23-03952Revisiting Flow Augmentation Bypass for Cerebrovascular Atherosclerotic Vaso-occlusive Disease: An Institutional Study and Review of The LiteraturePLOS ONE

Dear Dr. Abdelgadir,

Thank you for submitting your manuscript to PLOS ONE. After careful consideration, we feel that it has merit but does not fully meet PLOS ONE’s publication criteria as it currently stands. Therefore, we invite you to submit a revised version of the manuscript that addresses the points raised during the review process.

All reviewers are concerned with small number and its related recurrence/survival rate, as compared to previous larger RCT. I realize they have been addressed in the limitation section; however the authors should understand if it is methodologically acceptable or not (e.g. sample size to derive suitable statistical power). 

We look forward to receiving your revised manuscript.

Kind regards,

Tatsushi Mutoh

Academic Editor

PLOS ONE

Journal Requirements:

Reviewers' comments:

Reviewer's Responses to Questions

**Comments to the Author**

1. Is the manuscript technically sound, and do the data support the conclusions?

Reviewer #1: Yes

Reviewer #2: Partly

Reviewer #3: Yes

Reviewer #4: Yes

2. Has the statistical analysis been performed appropriately and rigorously? 

Reviewer #1: Yes

Reviewer #2: N/A

Reviewer #3: Yes

Reviewer #4: Yes

3. Have the authors made all data underlying the findings in their manuscript fully available?

Reviewer #1: Yes

Reviewer #2: Yes

Reviewer #3: Yes

Reviewer #4: Yes

4. Is the manuscript presented in an intelligible fashion and written in standard English?

Reviewer #1: Yes

Reviewer #2: Yes

Reviewer #3: Yes

Reviewer #4: Yes

5. Review Comments to the Author

Reviewer #1: The authors present a single-institutional retrospective review of patients receiving STA-MCA bypass from 2013 to 2021 for atherosclerotic steno-occlusive disease. The primary outcome evaluated was time to post-operative stroke. A total of 20 patients met inclusion criteria (2.5/year). Only one patient (5%) developed a stroke at approximately 2 months post op. A total of 3 seizures and one infection occurred. They showed an improvement in mRS as well as continued patency of all 20 bypasses.

Specific Comments:

• The authors should clearly mention that improvement in mRS likely has a contribution to recovery from the patient’s initial stroke.

• The authors should describe how many direct bypasses are being done for other conditions during this time-period, 2.5/year is a low amount to be considered a higher-volume center for these even if the results are good.

o Is this low number due to a strict selection criterion or are others in the group performing surgery as well? One would assume in a high-volume tertiary referral academic center (based on the selection criteria described) that this number would be higher.

• The median follow-up time is less than 1 year in an 8-year series, why is this?

o If the follow up is so short, how can you know these patients did not have a stroke or disability in the long term? This should be clearly spelled out in the limitations of this study.

• The authors should reference the recent systematic review in their comparison to published studies. There are several more articles than the 8 studies they mention that are recent and relevant to this manuscript. 1

• How was bypass patency assessed?

Overall Assessment:

This is a small retrospective study that does not show novel findings. However, the authors correctly conclude (in our opinion) that there is a patient population that still can benefit from revascularization EC-IC surgery in the modern era. The patient’s receiving surgery now typically include patients with refractory watershed strokes (which differs from COSS). Additionally, there continues to be improvement in imaging, surgical technique, and postoperative care. The 14.4% perioperative stroke rate in COSS is significantly different than what is presented in this study. This article should be accepted once the revisions above are addressed.

References:

1. Nguyen VN, Motiwala M, Parikh K, et al. Extracranial-Intracranial (EC-IC) Cerebral Revascularization for Atherosclerotic Vessel Occlusion: An Updated Systematic Review of the Literature. World Neurosurg. Feb 7 2023;doi:10.1016/j.wneu.2023.02.003

Reviewer #2: The authors retrospectively reviewed previous STA-MCA bypasses performed at their institution. Patients with recurrent or advanced stroke were evaluated, in part by CT perfusion studies. They performed an excellent procedure with an average blockade time of less than 15 minutes and a 100% patency rate; of the 20 patients, one (5%) had a recurrence during the observation period, but they stated that STA-MCA bypass is an effective method for recurrent strokes.

The results of the EC-IC bypass trial by Barnett et al. led to numerous criticisms by many neurosurgeons that EC-IC bypass is effective if the indications are not wrong, or that there were technical problems in the surgical group. This criticism arose from the impression that recurrent cerebral infarction could be prevented in the form of this paper.

Since this impressionistic criticism did not go far enough, the JET study and the COSS study were conducted to investigate a limited number of patients by accurately diagnosing the pathophysiology, evaluating cerebral blood flow using PET and SPECT, and assessing reserve capacity.

The authors' study is a case series without a control group, and the time since the first ischemic attack varies; the control patients are different from those in Abdurauf et al.'s acute bypass for progressive stroke. Although CT perfusion was performed in some patients, it was not an adequate assessment of blood flow. In addition, ischemia in the ACA and PCA regions is included, not only ischemia in the MCA region. It is unclear whether these were the result of evaluation of collateral blood flow by cerebral angiography or other means, and whether blood flow could be provided by STA-MCA bypass, which does not increase blood flow in the arteries in the direct perfusion zone.

The recurrence of one of the 20 patients is not an event that would be indicated by a survival curve, and again, in the absence of a control group, it can only be explained allegorically.

Reviewer #3: The authors present a retrospective, single-center review of patients undergoing EC/IC bypass for non-moyamoya vaso-occlusive disease that is refractory to medical therapy. They report reasonable outcomes that are similar to prior reports. They provide a discussion of the relevant literature and conclude that referral for EC/IC bypass in this patient population is a reasonable option and may be beneficial.

The article has some minor errors in grammar but is otherwise well written. The authors report similar numbers of cases to the cited literature and do not seem to add anything novel to the current fund of knowledge. Multiple centers have reported their experience and it is largely similar. In this reviewer's opinion the addition of another small series does not add anything significant to the current literature.

Reviewer #4: The manuscript is technically sound and data support the conclusions.

Statistical analysis has been performed appropriately.

The authors made all data in their manuscript fully available.

The manuscript is presented in an intelligible fashion and written in standard English.

I congratulate authors for sharing their experience on this retrospective cohort of patients receiving STA-MCA bypass in atherosclerotic vaso-occlusive disease (VOD).

I fully agree with their conclusion that STA-MCA bypass should remain an option for patients with VOD with recurrent strokes who fail medical therapy.

I would appreciate if authors expand on the surgical details regarding those patients with complications:

1.-One patient with postoperative scalp infection: Was this patient the one requiring double barrel bypass? What is the average size of the incision? Was a linear incision performed to harvest parietal branch of STA? Or curvilinear incision for frontal branch of STA? Or was a Y-type of incision performed?

2.-For the patient with one stroke, or the three with seizures: How was the management of the micro-cortical vessels? Where those vessels electrocoagulated to allow mobilization of the recipient M4 vessel? (Sharafeddin F, Lopez-Gonzalez MA. Micro-clipping of small cortical branches during extracranial to intracranial bypass for complete cortical blood supply preservation. Neurochirurgie. 2022 Oct;68(5):546-549. doi: 10.1016/j.neuchi.2022.02.005. Epub 2022 Mar 7. PMID: 35272857).

6. PLOS authors have the option to publish the peer review history of their article (what does this mean?). If published, this will include your full peer review and any attached files.

Reviewer #1: No

Reviewer #2: **Yes: **Toshikazu Kimura

Reviewer #3: **Yes: **Jonathan Russin

Reviewer #4: **Yes: **Miguel Angel Lopez-Gonzalez, MD

---

## [Author Response · Author response to Decision Letter 0]

5 May 2023

Response to Reviewers

Thank you for submitting your manuscript to PLOS ONE. After careful consideration, we feel that it has merit but does not fully meet PLOS ONE’s publication criteria as it currently stands. Therefore, we invite you to submit a revised version of the manuscript that addresses the points raised during the review process.

We thank the editor and the reviewers for the opportunity to submit a revised version of our manuscript. We have re-formatted the article and figures to meet PLOS ONE guidelines. The minimal data set underlying the study is included as supporting file S1. The corresponding author has validated their ORCID iD. 

All reviewers are concerned with small number and its related recurrence/survival rate, as compared to previous larger RCT. I realize they have been addressed in the limitation section; however, the authors should understand if it is methodologically acceptable or not (e.g., sample size to derive suitable statistical power). 

Thank you for raising this important point. As we have highlighted in our manuscript, both the EC-IC Bypass and COSS trials were landmark studies not only due to their findings, but also due to overcoming the significant difficulty of conducting randomized clinical trials in a surgical subspecialty. However, as suggested by the large study by Wessels et al., there is a subset of patients who are very high-risk for recurrent disease that may derive benefit from the procedure. Owing to rarity and the necessity of intervention in these patients, randomization in a new study may be difficult. Thus, a descriptive analysis of our experience with these high-risk patients is an important piece of a growing body of evidence that may eventually be practice changing for neurosurgery as a whole. Accordingly, we have edited our limitations section to read: 

Nonetheless, this study represents an important contribution to the growing body of post-COSS literature of bypass for VOD following failure of medical therapy in high-risk stroke subtypes.

Reviewer #1: The authors present a single-institutional retrospective review of patients receiving STA-MCA bypass from 2013 to 2021 for atherosclerotic steno-occlusive disease. The primary outcome evaluated was time to post-operative stroke. A total of 20 patients met inclusion criteria (2.5/year). Only one patient (5%) developed a stroke at approximately 2 months post op. A total of 3 seizures and one infection occurred. They showed an improvement in mRS as well as continued patency of all 20 bypasses.

Specific Comments:

• The authors should clearly mention that improvement in mRS likely has a contribution to recovery from the patient’s initial stroke.

We appreciate the opportunity to correct this oversight. We have addressed this in the discussion section of our article, which now reads:

While single-point changes in the mRS score are clinically relevant, the seven-level ordinal scale has been subjected to dichotomous analyses throughout its use as an endpoint in stroke RCTs; further, repeated measures of the mRS score highlight a general increase due to post-stroke recovery irrespective of treatment. 

In addition, we have cited the recent study by Chye et al. highlighting this observation [1].

[1] Chye A, Hackett ML, Hankey GJ, et al. Repeated Measures of Modified Rankin Scale Scores to Assess Functional Recovery From Stroke: AFFINITY Study Findings. J Am Heart Assoc. 2022;11(16):e025425. doi:10.1161/JAHA.121.025425

• The authors should describe how many direct bypasses are being done for other conditions during this time-period, 2.5/year is a low amount to be considered a higher-volume center for these even if the results are good.

We thank Reviewer #1 for raising this important comment. During the study period, 79 total EC-IC bypasses were performed by the senior author (A.R.Z). 39 of these were for Moyamoya disease; an additional 20 received EC-IC bypass for aneurysm. The remainder are represented in the study. To better indicate our experience with these procedures, we have added the following to the Results section, subsection “Patient clinical and surgical characteristics”:

During the study period, 79 patients were treated with EC-IC bypass. Of those excluded, 39 (49.4%) received bypass for Moyamoya disease and 20 (25.3%) for aneurysm or vertebrobasilar insufficiency. The remaining 20 patients received open vascular surgery for atherosclerotic VOD.

We have also edited the methods section to read as follows:

Patients who were undergoing STA-MCA bypass for Moyamoya disease, vertebrobasilar insufficiency, or aneurysm were excluded from the study. Other donor-recipient vessel pairs and bypasses for tumors were similarly excluded.

o Is this low number due to a strict selection criterion or are others in the group performing surgery as well? One would assume in a high-volume tertiary referral academic center (based on the selection criteria described) that this number would be higher.

We believe the responses above should clarify this question – the majority of EC-IC bypass cases performed by the senior author during the study period were not for the indication of vaso-occlusive disease. The reviewer’s assumption regarding a multi-surgeon setting is correct: we conducted this study as a single-surgeon series to mitigate differences in technique and experience.

• The median follow-up time is less than 1 year in an 8-year series, why is this?

Thank you for highlighting this important point. The median follow-up duration in the study was 7.7 months (range 1.2 – 43.7 months). As discussed above and in the manuscript, this study was conducted at a tertiary referral center with a large catchment area. Due to factors specific to this population, including a high attrition rate, local follow-up is typically pursued following initial post-operative clinic visits in patients without major complications.

o If the follow up is so short, how can you know these patients did not have a stroke or disability in the long term? This should be clearly spelled out in the limitations of this study.

We are grateful for the opportunity to improve our manuscript and agree that this limited follow-up is a limitation of our study. In keeping with our above response, we have edited the Limitations section in the Discussion to read the following:

The median follow-up duration in this study was 7.7 (1.2 – 43.7) months. This study was conducted at a tertiary referral center with a large catchment area, with a combination of location and patient specific factors resulting in limited follow-up.

It is our hope that this response has satisfied the relevant questions.

• The authors should reference the recent systematic review in their comparison to published studies. There are several more articles than the 8 studies they mention that are recent and relevant to this manuscript. [1]

We thank Reviewer #1 for highlighting this important systematic review, which was not published at the time of initial submission. While our review was not systematic in nature, we only included articles in which all patients were treated post-COSS; articles published after 2011 that had overlapping times of treatment were not included. Several of the articles termed “post-COSS” by Nguyen et al. do not meet this criterion. Nonetheless, we have incorporated the conclusions of this article into the Discussion as follows:

Indeed, a recent systematic review of EC-IC bypass for VOD suggested a trend towards decreased perioperative (5.7%) and overall stroke (9.1%) rates over time.(25)

• How was bypass patency assessed?

The authors apologize for this important oversight and are thankful for the opportunity to correct it. We have edited the Methods section, subsection “Patient selection and operation” as follows:

Bypass patency is confirmed intra-operatively with indocyanine green videoangiography and post-operatively via CT angiogram.

Overall Assessment:

This is a small retrospective study that does not show novel findings. However, the authors correctly conclude (in our opinion) that there is a patient population that still can benefit from revascularization EC-IC surgery in the modern era. The patient’s receiving surgery now typically include patients with refractory watershed strokes (which differs from COSS). Additionally, there continues to be improvement in imaging, surgical technique, and postoperative care. The 14.4% perioperative stroke rate in COSS is significantly different than what is presented in this study. This article should be accepted once the revisions above are addressed.

Many thanks to Reviewer #1 for their favorable assessment of our study. 

References:

1. Nguyen VN, Motiwala M, Parikh K, et al. Extracranial-Intracranial (EC-IC) Cerebral Revascularization for Atherosclerotic Vessel Occlusion: An Updated Systematic Review of the Literature. World Neurosurg. Feb 7 2023;doi:10.1016/j.wneu.2023.02.003

Reviewer #2: The authors retrospectively reviewed previous STA-MCA bypasses performed at their institution. Patients with recurrent or advanced stroke were evaluated, in part by CT perfusion studies. They performed an excellent procedure with an average blockade time of less than 15 minutes and a 100% patency rate; of the 20 patients, one (5%) had a recurrence during the observation period, but they stated that STA-MCA bypass is an effective method for recurrent strokes.

The results of the EC-IC bypass trial by Barnett et al. led to numerous criticisms by many neurosurgeons that EC-IC bypass is effective if the indications are not wrong, or that there were technical problems in the surgical group. This criticism arose from the impression that recurrent cerebral infarction could be prevented in the form of this paper.

Since this impressionistic criticism did not go far enough, the JET study and the COSS study were conducted to investigate a limited number of patients by accurately diagnosing the pathophysiology, evaluating cerebral blood flow using PET and SPECT, and assessing reserve capacity.

The authors' study is a case series without a control group, and the time since the first ischemic attack varies; the control patients are different from those in Abdurauf et al.'s acute bypass for progressive stroke. Although CT perfusion was performed in some patients, it was not an adequate assessment of blood flow. In addition, ischemia in the ACA and PCA regions is included, not only ischemia in the MCA region. It is unclear whether these were the result of evaluation of collateral blood flow by cerebral angiography or other means, and whether blood flow could be provided by STA-MCA bypass, which does not increase blood flow in the arteries in the direct perfusion zone.

The recurrence of one of the 20 patients is not an event that would be indicated by a survival curve, and again, in the absence of a control group, it can only be explained allegorically. 

We thank Reviewer #2 for their favorable assessment of our article and accurate summation of the literature that led us to conduct the present study. We have attempted to recognize the limitations correctly assessed by this reviewer in our article and hope that this is sufficient. All decisions regarding statistical description and data representation made in this article were conducted in collaboration with two biostatisticians. 

Reviewer #3: The authors present a retrospective, single-center review of patients undergoing EC/IC bypass for non-Moyamoya vaso-occlusive disease that is refractory to medical therapy. They report reasonable outcomes that are similar to prior reports. They provide a discussion of the relevant literature and conclude that referral for EC/IC bypass in this patient population is a reasonable option and may be beneficial.

The article has some minor errors in grammar but is otherwise well written. The authors report similar numbers of cases to the cited literature and do not seem to add anything novel to the current fund of knowledge. Multiple centers have reported their experience and it is largely similar. In this reviewer's opinion the addition of another small series does not add anything significant to the current literature.

We greatly appreciate Reviewer #3’s attention to our article. As highlighted in our responses above, we believe that experiences with this surgical technique in high-risk populations may form the basis of practice given the challenges to randomization in such a patient cohort. Additionally, we have attempted to correct any grammatical errors noted by the reviewer. We thank the reviewer for the opportunity to further strengthen our article.

Reviewer #4: The manuscript is technically sound, and data support the conclusions.

Statistical analysis has been performed appropriately.

The authors made all data in their manuscript fully available.

The manuscript is presented in an intelligible fashion and written in standard English.

I congratulate authors for sharing their experience on this retrospective cohort of patients receiving STA-MCA bypass in atherosclerotic vaso-occlusive disease (VOD).

I fully agree with their conclusion that STA-MCA bypass should remain an option for patients with VOD with recurrent strokes who fail medical therapy.

The authors are greatly appreciative of Reviewer #4’s favorable assessment of our article and are in agreement regarding the utility of this technique.

I would appreciate if authors expand on the surgical details regarding those patients with complications:

1.-One patient with postoperative scalp infection: Was this patient the one requiring double barrel bypass? What is the average size of the incision? Was a linear incision performed to harvest parietal branch of STA? Or curvilinear incision for frontal branch of STA? Or was a Y-type of incision performed?

We appreciate the opportunity to describe this complication; we did not previously include the details as we felt they were irrelevant to the present study. This patient underwent single barrel bypass. They underwent a groin procedure during hospitalization and presented post-operatively with drainage from scalp and groin wounds. Groin cultures grew MRSA, and this resolved following IV vancomycin treatment.

Regarding general incision technique, the senior author’s typical practice is to conduct Doppler ultrasound to identify the path of the superficial temporal artery. The resultant linear or curvilinear incision is made following the artery to expose 8 – 10 cm of the donor vessel only, with minimal additional dissection to expose the parietal branch of the STA. We have included this in the manuscript Methods section, subsection “Patient selection and operation” as the following:

Once in the operating room, the senior author’s typical practice is to conduct Doppler ultrasound to identify the path of the superficial temporal artery. The resultant linear or curvilinear incision is made following the artery to expose 8 – 10 cm of the donor vessel only, with minimal additional dissection to expose the parietal branch of the STA. 

2.-For the patient with one stroke, or the three with seizures: How was the management of the micro-cortical vessels? Where those vessels electrocoagulated to allow mobilization of the recipient M4 vessel? (Sharafeddin F, Lopez-Gonzalez MA. Micro-clipping of small cortical branches during extracranial to intracranial bypass for complete cortical blood supply preservation. Neurochirurgie. 2022 Oct;68(5):546-549. doi: 10.1016/j.neuchi.2022.02.005. Epub 2022 Mar 7. PMID: 35272857).

In the senior author’s practice, electrocoagulation of the micro-cortical vessels is typically avoided. A recipient vessel is identified that does not require manipulation or sacrifice of micro-cortical vessels. This is included in the Methods section, subsection “Patient selection and operation” as the following:

A recipient site requiring no sacrifice or manipulation of microcortical vessels is selected.

The study authors thank the editor and reviewers for their time taken to improve this manuscript. It is our hope that the changes made have greatly improved the quality of the article and that it is found suitable for publication in this journal. 

Sincerely,

The authors

---

## [Editor Report · Decision Letter 1]

7 May 2023

Revisiting flow augmentation bypass for cerebrovascular atherosclerotic vaso-occlusive disease: Single-surgeon series and review of the literature

PONE-D-23-03952R1

Dear Dr. Abdelgadir,

We’re pleased to inform you that your manuscript has been judged scientifically suitable for publication and will be formally accepted for publication once it meets all outstanding technical requirements.

Kind regards,

Tatsushi Mutoh

Academic Editor

PLOS ONE
---

## [Editor Report · Acceptance letter]

10 May 2023

PONE-D-23-03952R1 

Revisiting flow augmentation bypass for cerebrovascular atherosclerotic vaso-occlusive disease: Single-surgeon series and review of the literature 

Dear Dr. Abdelgadir:

I'm pleased to inform you that your manuscript has been deemed suitable for publication in PLOS ONE. Congratulations! Your manuscript is now with our production department. 

Kind regards, 

on behalf of

Dr. Tatsushi Mutoh 

Academic Editor

PLOS ONE